# Experience of Pharmacists with Anti-Cancer Medicine Shortages in Pakistan: Results of a Qualitative Study

**DOI:** 10.3390/ijerph192316373

**Published:** 2022-12-06

**Authors:** Sundus Shukar, Fatima Zahoor, Sumaira Omer, Sundas Ejaz Awan, Caijun Yang, Yu Fang

**Affiliations:** 1Department of Pharmacy Administration and Clinical Pharmacy, School of Pharmacy, Xi’an Jiaotong University, Xi’an 710061, China; 2Jeffrey Cheah School of Medicine and Health Sciences, Monash University, Bandar Sunway 47500, Malaysia; 3Faculty of Pharmacy, Gadjah Mada University, Yogyakarta 55281, Indonesia

**Keywords:** anti-cancer, medicine shortages, oncology medicine shortages, essential medicine shortages, chemotherapeutic shortages, Pakistan

## Abstract

This study aimed to examine the current situation of anti-cancer drug shortages in Pakistan, namely its determinants, impacts, adopted mitigation strategies, and proposed solutions. Qualitative semi-structured, in-depth interviews were conducted with 25 pharmacists in oncology hospitals in Pakistan from August to October 2021. Data were collected in person and online, recorded, and subjected to inductive thematic analysis after being transcribed verbatim. Most participants experienced anti-cancer drug shortages that increased during the pandemic. Etoposide, paclitaxel, vincristine, dacarbazine, and methotrexate were frequently short. Important causes included the compromised role of regulatory authorities, lack of local production, and inventory mismanagement. The impacts were delayed/suboptimal treatment and out-of-pocket costs for patients, patients’ prioritization, increased workload, negative work environment, and patients’ trust issues for pharmacists. The participants proposed that a cautious regulator’s role is needed to revise policies for all stakeholders and support all stakeholders financially at their level to increase access to these medicines. Based on the outcomes, it is clear that anti-cancer medicine shortages are a current issue in Pakistan. Governmental authorities need to play a role in revising policies for all levels of the drug supply chain and promoting local production of these drugs. Stakeholders should also collaborate and manage inventory.

## 1. Introduction

According to the World Health Organization (WHO), access to medications is a ubiquitous and fundamental right for individuals to achieve good health standards [1], and shortages of these drugs are a global challenge influencing numerous medicines in diverse therapeutic areas. These shortages have been influencing cancer treatment that is comprised of agents with costly drugs having a low therapeutic index [2], few alternative options, and agents being used in combination. These distinguishing features have led to severe shortage impacts [3,4,5,6,7,8]. Moreover, most countries import these expensive agents [4], leading to higher economic costs for patients and institutions, but the ultimate burden is on patients [7]. The situation is more critical for pediatric cancer patients [9]. Chemotherapeutic medicines are one of the top five most affected drug classes [9], commonly short in developed and developing countries and putting an enormous burden on patients and healthcare organizations [10]. The condition is more severe in developing regions as fewer drugs are available, with a lack of cancer-related data and increased morbidity and mortality [11].

Pakistan, a South Asian developing country with accelerating cancer cases, is ranked 154th for health system quality [12]. The private sector attends almost 70.0% of the population [12]. The country is a big pharmaceutical hub but imports most of its cancer drugs from other countries. With the increasing cancer burden [13,14], absence of a national cancer registry/drug shortage reporting system [15,16], the compromised role of drug regulatory authorities, outdated drug policies [17], inadequate treatment facilities [18], few research studies [4,17], and low availability, the shortages of these drugs further make this issue challenging [17,19]. A study conducted in a tertiary care hospital also found a severe shortage of anti-cancer drugs [17].

Addressing anti-cancer medicine shortages in low-income countries is a difficult task. An in-depth estimation is of the utmost need now to ensure optimal patient care and to tackle the expected mortality and morbidity of cancer. There is a lack of evidence-based research studies to analyze and handle this issue in Pakistan, so the study aimed to evaluate, characterize and assess this issue in Pakistan from the perspectives of pharmacists to deliver ideas to facilitate understanding of anti-cancer drug shortages in Pakistan, their unique nature and to encourage further analysis of policies that could reduce shortages. The study explored the current situation, its determinants, and impacts and adopted both mitigation strategies and recommendations.

## 2. Materials and Methods

### 2.1. Study Design and Data Collection

We designed a qualitative study to explore significant anti-cancer drug shortages in Pakistan.

A semi-structured interview guide was designed from a deep literature review and study objectives adopted from similar global studies [16,20,21,22]. Three investigators reviewed the interview guide. Then, a pilot study was conducted by interviewing three participants from different regions of the country. Based on the feedback from the academic review and pilot study, we modified and refined the interview guide. The three interviewees involved in the pilot study were excluded from the final study. The final form of the interview guide was comprised of five sections. Section one contained general information, section two contained general thoughts about anti-cancer medicines shortages, section three contained reasons of anti-cancer medicines shortages, section four contained impacts, and section five contained adopted mitigation strategies (Appendix A).

We focused on hospital pharmacists, as they faced anti-cancer drug shortages on the front line; involved in problem evaluation, management and knowledge sharing. They commonly deal with logistics and procurement, directly connected with manufacturers, distributors, suppliers and oncologists. In many hospitals, they act as an information source for other health professionals. Moreover, in Pakistan, access to pharmacists is feasible in comparison to other health professionals [23].

In the sampling process, we first prepared a list of cancer hospitals (general hospitals with oncology departments or specialized from all public/private sectors) throughout the country. Based on the availability of cancer hospitals, we divide the whole country into five regions (Punjab, Sindh, KPK, Islamabad, and Baluchistan). In each region, the number of pharmacists was selected depending on the population of hospital pharmacists/region, and finally, we invited a total of 30 pharmacists through email and/or phone.

All of the interviews were conducted from August to October 2021, face-to-face or online via Zoom, according to participants’ preference. English is the official language in Pakistan and is used throughout the country, so the interviews were conducted in English and audio-recorded.

### 2.2. Data Analysis

The audio-recorded interviews were transcribed and analyzed using NVivo through inductive thematic analysis. Data familiarization was acquired by studying the transcribed scripts several times. Firstly, two researchers manually coded the data. Relevant words, statements, and utterances reflecting the research objectives were annotated and initial inductive codes were constructed to partition the data into separately coded sections. Codes from the early interviews were utilized to create a coding schema to analyze later interviews. These emerged codes were used to analyze the data in NVivo. Focused coding was carried out after the initial coding. Focused coding involves the exploration of the relationship between initial codes depending on difference, similarity, sequence, frequency, causation, and correspondence. The finalized inductive codes were segmented into meaningful categories. To conceptualize the data, different categories were combined to generate themes and subthemes. Before developing final themes, transcripts, codes, and categories were recursively examined. To strengthen the reliability of the findings, quantification (recording the frequency per each code) and tabulation were used. Each participant’s response was quantified once, and the outcome suggested by the majority of participants was considered a significant finding. Regular group meetings were held by the research team to ensure that everyone had the same understanding and perspective of the developed categories.

### 2.3. Consent and Ethics Approval

All stakeholders gave their verbal consent to participate in the research. Prior to commencing the interviews, participants were asked to read the study’s purpose and the confidentiality declaration. Respondents also had the option of terminating their participation at any time. The study did not reveal the names of the respondents, and the audio recordings were saved properly.

## 3. Results

### 3.1. General Characteristics

Out of 30 pharmacists invited, 5 respondents (16.6% refusal rate) refused to participate in the research study due to their busy schedules and paucity of interest in the study. Of the 25 pharmacists interviewed, 17 (68.0%) were male and 8 (32.0%) were female. The interview duration ranged from 24 to 65 min, with an average duration of 35 min. The participants’ age ranged from 26 to 62 years, with an average age of 33 years (Table 1).

In this study, five key themes were retrieved, including dynamics and perception of anti-cancer medicines shortage, its determinants, impacts, adopted mitigation strategies, and future interventions.

### 3.2. Theme 1: Dynamics and Perception of Anti-Cancer Medicines Shortage

Almost all participants whether working in a governmental or private hospital, agreed that the oncology drug shortage is a current issue (Table 2). The frequency of shortages varied for them, the most recurrent answer was monthly (28%), followed by daily, weekly and quarterly (all 16%), and finally half yearly and yearly (both 12%). Most frequently short anti-cancer medicines included etoposide, paclitaxel, vincristine, dacarbazine, methotrexate, gemcitabine, vinblastine, and bleomycin. Participants declared that shortages have surged in recent years, particularly in the COVID-19 pandemic which aggravates the shortage and also led to increased prices for these expensive agents. Most pharmacists agreed that they preferred the patient on a first-come, first-get basis.

### 3.3. Theme 2: Determinants of Anti-Cancer Medicines Shortage

The interviewees were questioned from manufacturers up to the hospital level to determine the real causes of anti-cancer drug shortages in Pakistan. Of several reasons for the anti-cancer medicine shortages referenced by them, the most important reasons are ranked in a descending order: Compromised role of regulatory authority, lack of local production, and inappropriate inventory management (Table 3).

### 3.4. Theme 3: Impacts of Anti-Cancer Medicines Shortage

For clinical impacts on patients, pharmacists reported that anti-cancer medicine shortages led to delayed or suboptimal treatment due to the use of alternatives, ultimately leading to disease progression. These shortages caused increased out-of-pocket-cost for patients in all ways either arranging the expensive alternative or delaying the treatment causing increased hospitalization, traveling costs (to arrange medicine) and disease progression. In government and semi-government hospitals, the medicines are available at affordable prices, but in case of shortages, patients have to purchase from outside, affecting their treatment adherence.

The most promising impact on the pharmacists reported by most of them was increased workload. They also reported the negative work environment as these shortages questioned their competency. Moreover, they lost the trust of patients and become frustrated. Patients blamed them for not providing them desired medicine (Table 4).

### 3.5. Theme 4: Adopted Mitigation Strategies

The pharmacists shared proactive and counteractive actions to mitigate anti-cancer medicine shortages. The most frequent proactive action reported by several pharmacists was managing the inventory appropriately. Many private hospitals have electronic inventory systems that helped them to manage inventory as reported by some pharmacists. The counteractive actions included managing within a hospital or from external sources. Pharmacists reported that they managed shortages by contracting other hospitals, suppliers, or even the black market if there is no option available. Another most frequently mentioned counteractive adopted strategy was importing either patient need base or institution need base (Table 5).

### 3.6. Theme 5: Future Interventions for Anti-Cancer Medicines Shortage

For future interventions, almost all participants agreed to have a vigilant role from regulators to updated policies for anti-cancer drug registration, import, license renewal, fixing drug prices and profit margins (dollar fluctuation causes loss of profit). Moreover, governmental authorities should support manufacturers for quality local production and other stakeholders financially (Table 6).

## 4. Discussion

In developing countries, the typology of anti-cancer medicine shortages needs to be explored and studied [24]. The accessible health professionals (hospital pharmacists) admitted the presence of anti-cancer medicine shortages in their hospital but the frequency of shortages varies. The reason is the varied financial stability of institutions. A survey conducted in Saudi Arabia also reported the presence of oncology drug shortages with similar statistics [23].

Participants declared that shortages have surged and aggravated in recent years in the COVID-19 pandemic leading to increased prices for these expensive agents. A study on the impact of COVID-19 on global drug shortages emphasized a similar aspect [25]. Studies in the US also highlighted the increase in medicine shortages during the pandemic [26,27].

In Pakistan, the most prominent cause of anti-cancer drug shortages is the compromised role of the drug regulatory authorities. This has led to outdated policies for drug registration, import, license renewal, fixed drug price, and profit margin (dollar fluctuations). Hence, suppliers are hesitant to import these essential molecules. The compromised role also leads to issues in the hand of other stakeholders, for example, unfair distribution, stocking (selling short expiry drugs/increased prices), increased contracts than capacity, and purchasing from the gray market (compromised efficacy drugs) by distributors [28]. According to a study in Ghana, the compromised role of the drug regulatory authority is the main reason for the lack of access and shortages of necessary medicines in LMICs [29]. A recent study from Pakistan also highlighted regulatory issues as the prominent cause of shortage [17]. Some causes related to other stakeholders are the presence of a single supplier/drug, lack of resources, lack of communication among stakeholders, and small market size. However, these issues could also be solved by the regulatory authorities. Another study in Pakistan mentioned similar causes for drug shortages [16]. Many participants stated that communication among stakeholders within the same institution and among different institutions is very important for handling the anti-cancer drug shortages smoothly [4]. This cooperation leads to an improvement in operational transparency. A study in Saudi Arabia also agreed that a lack of communication aggravated drug shortages [30].

Another important cause is the inappropriate inventory management for raw material at the manufacturer’s hands and anti-cancer drugs at the wholesalers’/hospitals’ which led to shortages and the reason could be the mismanagement of the procurement department (absence of competent pharmacists) or just-in-time (JIT) inventory due to budget constraints (absence of support from governmental authority) [24]. A study conducted at the public hospitals of South Africa stated that continuous monitoring and computerized system played an important role in drug shortage management [31].

The impacts of anti-cancer drugs shortage in Pakistan, a low-income country with the absence of a national cancer registry/drug shortage platform [17], increasing cancer cases and fewer registered drugs [17], are underestimated. Most participants highlighted suboptimal treatment and delayed treatment along with increased out-of-pocket costs. Similar results are found in studies from the US, Egypt [32] and Morocco [33]. The results are also consistent with another recent study carried out in Pakistan [32] and the one carried out in the US where 65.0% of the participants mentioned delayed treatment due to shortage [16,34]. A survey of the drug shortages’ impact on acute care institutions in the US also stated that shortages led to delayed treatment (adverse drug interactions) and some drug shortages impacts are expected to be not reported fully [35]. Studies throughout the globe agree that medicines shortages lead to out-of-pocket costs and if the condition is cancer, then this burden will be huge [36]. In government hospitals, medicines are available at affordable cost, and if these medicines are short, patients have to purchase expensive medicines or medicines from a gray market with compromised quality. Ultimately, patients/caregivers become hopeless and frustrated [36]. Some participants also reported the need for an established and competent role of the pharmacist to handle this situation. Similar results were seen in a survey in Pakistan [32].

From the adopted mitigation strategies for anti-cancer drug shortages, it seems that the professionals are handling the situation well. However, in reality, it is an extremely challenging, time-consuming, and severe situation since both proactive and counteractive strategies are affected by outdated regulatory policies. In the case of proactive activities, even after well anticipation, financial constraints limit the procurement capacity, and only a few private setups or well-established NGO-based hospitals could be able to procure up to their need and for procuring more brands, only a few generics/brands are approved. For counteractive strategies, managing within hospitals leads to a change of protocol (rescheduled expensive treatment with or without compromised outcomes) [28], delayed treatment, and switching to alternatives (expensive second or third-line agents with less efficacy increased chances of medication error), all three would impact the optimum health outcomes to some extent [27,33,37].

Managing within the country would be challenging since few suppliers supply the drugs throughout countries so most probably the short drug in one institution will be short in others too [17]. Moreover, pharmacists or patients themselves have to purchase drugs from the gray market, where the short drugs are expensive and have compromised efficacy. In this case, the out-of-pocket cost for the patient will increase more due to purchasing expensive drugs [33]. Furthermore, arranging through import is an expensive and lengthy process that lags the treatment duration of the patients who are already vulnerable to a chronic condition at a specific disease stage [17].

A prominent role is required from regulators through (a) introduction of updated policies (policies for registration/import/license renewal/fix drug prices/profit margin/generic prescribing system/drug supply chain management including fair distribution/punishment for breaching); (b) providing financial support to stakeholders (register reputable manufacturers for local manufacturing, contracts with more than one suppliers/drug, the increase financial budget of oncology drugs for hospitals); (c) establishing cancer registry and national level drug shortage platform that would also work as a communication platform; (d) implementing research studies to get actual facts. Other stakeholders should play their role (i.e., manufacturers need to produce quality products/APIs and make a committee to deal with APIs shortages). Distributors should prefer fair distribution and control profit margins. Hospitals need to admit patients depending on their capacity and available stock, develop a drug shortage platform within hospitals, and give full play to the role of the pharmacist; tele-pharmacy can also contribute to those hospitals’ lack of competent pharmacists in this aspect of choosing suitable alternative medicines [38].

The study is the first of its kind in Pakistan to highlight the experience of pharmacists with anti-cancer medicines shortages through interviews and put forward some recommendations along with some highlighted research gaps. There are several limitations. Firstly, only pharmacists were included. In the future, additional research studies could be conducted to investigate the perceptions of other key stakeholders on anti-cancer medicine shortages. Secondly, there is a possible risk of bias and discrepancy in findings allied with using both online zoom meetings and face-to-face interviews. The study was designed for online zoom meetings in the pandemic era but few pharmacists were hesitant to participate online. For both types of participants, all details including the interview guide schema were shared through email, and concerning queries were answered to minimize potential biases.

## 5. Conclusions

Anti-cancer medicine shortages are a current issue in the health care system of Pakistan. The prominent causes of this are a lack of local production and the compromised role of regulatory authorities which have led to a deformed drug supply chain. The impacts of these shortages are underestimated due to the need for more data. The professionals tried to manage these shortages, but the role of governmental authorities is needed for mitigation. These authorities should develop policies for all levels of the drug supply chain, support the manufacturers for local production of anti-cancer drugs and their raw materials, and assist other stakeholders financially to increase access. Moreover, increasing communications among stakeholders, managing inventory appropriately, and increasing cancer research studies are also recommended.

## Figures and Tables

**Table 1 ijerph-19-16373-t001:** Response rate per province and the characteristics of interviewees, *n* (%).

Facility Location	Facility Type				
	Government Hospital	Semi-Government Hospital	Private Hospital	Trust	Total
Punjab	1 (50.0)	3 (50.0)	3 (50.0)	3 (27.0)	10 (40.0)
Sindh	1 (50.0)	1 (16.7)	3 (50.0)	2 (18.0)	7 (28.0)
KPK	0 (0.0)	1 (16.6)	0 (0.0)	5 (46.0)	6 (24.0)
Baluchistan	0 (0.0)	1 (16.7)	0 (0.0)	1 (9.0)	2 (8.0)
Total	2 (8.0)	6 (24.0)	6 (24.0)	11 (44.0)	25 (100.0)
**Demographic**			**Hospital pharmacists (*n* = 25)**		
Sex (*n*, %)					
Male			17 (68.0)		
Female			8 (32.0)		
Age (*n*, %)					
<28 years			8 (32.0)		
29–33 years			9 (36.0)		
34–38 years			4 (16.0)		
39–43 years			1 (4.0)		
44–48 years			3 (12.0)		
Oncology related work experience (mean ± SD)			5.9 ± 1.1		

**Table 2 ijerph-19-16373-t002:** Theme 1: Dynamics and perception of anti-cancer medicines shortage.

Subthemes	Categories and Subcategories	Quotations
Current experience with anti-cancer medicines	Varied frequency of shortage	Some medicines get shot on daily basis, some of them on weekly basis and those medicines which have very low consumption, that are monthly or annually or biannually (Participant 7).
	Brand shortages	We experienced a bit of shortage of brands of chemo medicines after every two to three months (P11)
	Shortages depends upon types of medicines	It depends upon type of drug effected, either it is first line drug, or second line drug or some targeted therapy (P25).
Impact of COVID-19 on shortages	COVID-19 aggravated the anti-cancer medicines shortages	In COVID-19, logistics are involved and major shortages happened. The import clearances of single brand/drug also take enough time. Drug is present at Pakistani port but not available to the patient. So before COVID-19, it was monthly but in COVID-19, it is daily or weekly (P8).
	COVID-19 led to an increase in the price of short medicines	Many wholesalers or distributors or manufacturers created artificial shortages to increase the rates and prices of medicines and sold them at black rates. It was all for monetary benefits (P4).
Ethical dilemma	Pharmacists have to prioritize the patients (First come, first get; Curative care patients will be preferred on the palliative care/dying one; Distant patients are preferred on nearer one; Poor are preferred on rich)	If we have started chemo of one patient, we have to ensure that patient must receive complete cycle of chemotherapy. We will never start to the new patient since we have to ensure first that those patients we have already, must complete their cycles (P1).
	Other professionals make the decision (Physicians, Multidisciplinary team meeting)	Basically, our hospital used to develop criteria in the multidisciplinary team meeting that included physicians, pharmacists, consultants, radiologists, and other professionals also. They basically developed a criteria for the best use of the shortage drug (P21).

**Table 3 ijerph-19-16373-t003:** Theme 2: Determinants of anti-cancer medicines shortage.

Subthemes	Categories and Subcategories	Quotations
Regulatory issues	Compromised role of DRAP (In the aspects of registration, license renewal, import, price management and supply chain management)	One of the things is the challenging drug registration process, which is a very cumbersome process, it’s not a robust process. So, any new anti-cancer treatment which comes into the global options is very late registered within Pakistan (P13).
		The important cause is the DRAP, the authority is not working properly, unable to understand the issues. If DRAP will not work honestly, then everyone has to face issues of registration, import, expiry, and black market (P10).
	Lack of local production	These agents are not manufactured in Pakistan (P17).
	Brand prescribing trend	Pharmaceutical companies attract physicians just to promote their brands and physician have to prescribe the particular brand (P3).
Suppliers/Distributors issues	Lack of local or international vendors	They are not much interested and dealing in anti-cancer where they have to invest capital and do precautions. Only a few dealers or wholesalers are dealing with anti-cancer drugs (P5).
	Increased contracts with hospitals than capacity	They took orders from so many hospitals and were unable to provide the expected stock which lead to a shortage (P2).
	Short expiry drugs	If a drug has short expiry (as most anti-cancer drugs), then automatically, the drug has a short market time and delays caused by distributors may cause shortages and loss of drug (P10).
	Unfair distribution	The focus of distributors/wholesalers is upon the big cities or big hospitals for more profit, and this also causes a shortage in remote areas or distant areas (P9).
Manufacturing issues	Raw material issues (high cost and unavailability)	The raw material is of very high cost now, that’s why they fail to get appropriate price and try to avoid manufacturing (P1).
	Quality concerns of locally production	
Hospital issues	Clerical issues	In government hospitals, there is purchasing committee involving admin, accounts and this takes a lot of time before going to purchase a drug leading to shortage (P9).
	Not involved	The hospital itself didn’t play that much role since they completely rely on the distributor, on the manufacturing somehow (P17).
Common causes among stakeholders	Lack of communication	There is a lack of communication among the oncology Society of Pakistan, drug regulatory authority of Pakistan, and the Ministry of Pakistan (P6).
	Artificial shortage	The main reason related to distributors and manufacturers, is they want their drug on high pricing, they increase their profit margins creating a fake shortage. Sometimes, they create a fake shortage to consume their short expiry bag (P3).
	Inappropriate inventory management	Basically, distributors have not maintained their inventory as per the patient’s need or either they do not have sufficient data for arranging the store of medicine for particular timing. The same is the case with hospitals (P12).
	Presence of a single supplier for a single drug	This is the most critical thing that we have one drug and one distributor, which is always alarming since if a distributor has some personal reason for the business and if he drops a sale or import, the impact comes on the country (P13).
	Demand fluctuation and small market size	There are a certain number of institutions, which are just dealing with oncology. So it’s not a huge number that is prescribed by every second physician (P3).

**Table 4 ijerph-19-16373-t004:** Theme 3: Impacts.

Subthemes	Categories and Subcategories	Quotations
Clinical impact	Suboptimal treatment	When a patient does not get his medicine on time, it obviously deteriorates his health. There will be long hospitalization, there will be long-term timing of the treatment schedule and automatically there will be a sub-optimal health condition (P22).
	Delayed treatment	The process of importing is a little bit lengthy leading to delayed treatment (P13).
	Adverse drug reactions	Adverse effects may occur using alternative/other therapies. This is due to the fact that in clinical settings, doctors choose the medicine on the basis of the patient profile, for example, a cardiotoxic adriamycin cannot be given to a cardiac patient but if the other choice is eliminated, it will go (P12).
	Death	Every day delay leads to increase mortality (P5).
Financial impact	Increased out-of-pocket costs for the patient	Patients have to bear their traveling costs, emotional costs, they have to pay more, they have to pay if there is a shortage and they get medicine from another city and they have to administer in another city (P8).
Impact on pharmacist	Increased workload	We have to sometimes spend our whole day figuring out how to counter the shortages (P14).
	Pharmacists become frustrated (Disturb pharmacist-patient relationship, accused of incompetency)	Our department and our higher authorities blame us that we have to meet our needs. So we are in a tightrope that we are pulled on both sides, from the one end by patients and from the other end by higher authorities (P22).

**Table 5 ijerph-19-16373-t005:** Theme 4 Adopted mitigation strategies.

Subthemes	Categories and Subcategories	Quotations
Proactive measure	Anticipated demand and shortages as well	We have an electronic inventory control system through which we can assess what are the actual needs of the institution, and how much smooth supplies do we have (P13).
	Procure more than one brand	Mostly we have approved two brands in our formulary. On the shortage of one, we go for the other (P8).
	Shortened the procurement decision-makers	We shortened the procedure of purchasing drugs by shortening the decision-makers for purchasing (P25).
Counter active measure	Manage within hospital (Change the protocol, delayed the treatment, switch to alternatives, compounding)	Sometimes we have to switch to a therapeutic equivalence after a doctor’s consultation (P14).
	Manage within country (Contact other hospitals, arrange drugs from the black market, contacting multiple suppliers, patient arrange their own medicine)	First of all, when we faced a shortage of anti-cancer medicine, we contacted our friends from other institutes, So we contacted their departments, and then we contacted other pharmacies (P22).
		Many times patients arrange their medicines. Sometimes those drugs are not available in hospitals or at the distributor level but are available on the black market for the patient so patients have to purchase (P12).
	Manage through import (Patient need base import, institution need base import)	One alternative way is to import the drug instead of waiting for the trial period. So the first thing is the integration. The second thing we go for is the patient need basis import process. The third thing we go for is the institutional import process to overcome such drug shortages (P13).

**Table 6 ijerph-19-16373-t006:** Theme 5: Future Interventions.

Subthemes	Categories and Subcategories	Quotations
Regulators	Introduce updated policies (Policies for registration/import/license renewal/fix prices/profit margin/drug supply chain management/those who break the law)	DRAP authorities should update regulations on which they have to allow the timely import of medicines. DRAP should also counter-check drug registration and license renewal policies to make the process smooth (P14).
		There must be penalties for those who break the laws and hold the medicines for the sake of profits, those who produce artificial shortages, and those who sell drugs at higher prices (P4).
	Financial support to stakeholders (Register reputable manufacturers for local manufacturing, contracts with more than one suppliers/drug, increase financial budget of oncology drugs for hospitals)	The government should encourage the pharmaceutical manufacturer to have more manufacturing plants and give some subsidiary support. They can encourage the pharmaceutical industries to produce a good number of anti-cancer drugs. Theyu can even export as well if they have some quality standard pharmaceutical manufacturing plants (P13).
	National level drug shortage platform	There is also a need for a national-level drug shortage platform to tackle this situation (P5).
Manufacturers	Local manufacturing of quality anti-cancer drugs/APIs	The drug manufacturers can make a union so that they can ask lawmakers and the government, and pressurize them to make factories and produce raw material from our own country (Participant 22).
	Make a committee to deal with APIs shortages	Drug manufacturing companies should make some special type of team that deals with the active ingredient shortages and timely purchase (P15).
Distributors	Fair distribution(Avoid fake shortage)	Distributors cannot hold the stock on their own hands for their own benefits, but they’re doing it seriously. They should make it available for the patient, for the hospitals, for the other drugstores easily (P17).
	Minimize profit margin	Two things, the government should facilitate them, but at the same time, they should minimize the profit margin and ensure that those drugs are available in the market (P15).
Hospitals	Need of proactive measures (Generic prescribing in hospitals, admit patients/stock available, drug shortage platform, established role of pharmacist)	Brand deficiency can only be replaced with a generic prescribing (P15).
		There should be a proper drug shortage platform, committee, or type of drug product shortage team that also act as a help desk where pharmacist or doctor inform their complaint about drug shortages (P15).
Shared Interventions among stakeholders	Anticipate and maintain a good inventory system	The hospital supply chain management or procure department should understand their needs and manage inventory accordingly (P10).
	Communication at the hospital level is very important	They should do agreements and contacts with distributors and other stakeholders to keep the drug flow. Other staff should corporate with each other to fix the solution of the medicine shortages (Participant 16).
	Research surveys to get actual facts	The data available in Pakistan is too short or not available at any forum. There is no cancer registry platform at national level (P12).

## Data Availability

Not applicable.

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
