# Peer review of "Experience of Pharmacists with Anti-Cancer Medicine Shortages in Pakistan: Results of a Qualitative Study"

_ijerph, 2022, doi:10.3390/ijerph192316373_

Round 1
Reviewer 1 Report
Small edits needed. For example line 97 should read, "...recording were listened to many time to transcribe...".
Please re-write the lines 84-90. It is not clear if you had an additional 30 subjects or 30 was the total in the sample. (I do see in subsequent writings it is 30, but in the first description it is just a bit confusing).
Your study is very well done and you should be congratulated on a very fine study with some very, very important findings. I enjoyed reading it.
Author Response
Response to the Reviewer 1 Comments
Comment 1: Small edits needed. For example line 97 should read, "...recording were listened to many time to transcribe...".
Answer: Thanks for your valuable comment and suggestion. The respective line has been deleted as the previous sentence has related information.
“The audio-recorded interviews were transcribed and analyzed using NVivo through inductive thematic analysis.”
Comment 2: Please re-write the lines 84-90. It is not clear if you had an additional 30 subjects or 30 was the total in the sample. (I do see in subsequent writings it is 30, but in the first description it is just a bit confusing).
Answer: Thanks for your valuable comment. The respective lines has been corrected as below.
“In the sampling process, we first prepared a list of cancer hospitals (general hospitals with oncology departments or specialized from all public/private sectors) throughout the country. Based on the availability of cancer hospitals, we divide the whole country into 5 regions (Punjab, Sindh, KPK, Islamabad, and Baluchistan). In each region, the number of pharmacists was selected depending on the population of hospital pharmacists/region, and finally, we invited a total of 30 pharmacists through email and/or phone.”
Reviewer 2 Report
This is a useful paper documenting the problems of supply for anti cancer drugs in Pakistan. The informants were representative pharmacists from cancer hospitals across the country and there were a number of useful suggestions as to how things could be improved.
There were a number of language problems:
Line 67 "intense need of issue" might be better "to explore the significant anti cancer drug shortages'
line 68 "intense" maybe "thorough" or maybe just delete
line76 "reasons of anti-cancer medicines" does not make sense to me
line 77 "contained past mitigation" maybe mitigation strategies
line 79 delete"and"
Line 97 "were listened many times" just delete..."the recordings were transcribed" might be better
Line 148 "the most Important reasons' (Not imported)
Line 264 "the projecting role is required" does not make sense to me
Line 298 "to at their level" needs deleting
Author Response
Response to the Reviewer 2
Thanks for your valuable comments and suggestions.
Comment 1: Line 67 "intense need of issue" might be better "to explore the significant anti-cancer drug shortages'. line 68 "intense" maybe "thorough" or maybe just delete
Answer: Thanks for your valuable comment and suggestion. Line 67 has been improved as below.
“We designed a qualitative study to explore the significant anti-cancer drug shortages in Pakistan.”
Comment 2: line76 "reasons of anti-cancer medicines" does not make sense to me. Line 77 "contained past mitigation" maybe mitigation strategies
Answer: Thanks for your valuable comment and suggestion. The respective line 76 has been improved with the addition of “shortages”, and line 77 has been improved with the addition of “adopted mitigation strategies” as given below.
“Section one contained general information, section two contained general thoughts about anti-cancer medicines shortages, section three contained reasons of anti-cancer medicines shortages, section four contained impacts, and section five contained adopted mitigation strategies.”
Comment 3: line 79 delete"and"
Answer: Thanks for your valuable comments. The respective line 79 has been improved with the replacement of “and” with “;” as two sentences were combined.
“We focused on hospital pharmacists, as they faced anti-cancer drug shortages on the front line; involved in problem evaluation, management and knowledge sharing.”
Comment 4: Line 97 "were listened many times" just delete..."the recordings were transcribed" might be better
Answer: Thanks for your valuable comments. Line 97 has been deleted
Comment 5: Line 148 "the most Important reasons' (Not imported)
Answer: Thanks for your valuable comments. Line 148 has been improved with the correction of a spelling mistake.
“Out of several reasons for the anti-cancer medicine shortages referenced by them, the most important reasons are ranked in descending order:”
Comment 6: Line 264 "the projecting role is required" does not make sense to me
Answer: Thanks for your valuable comments. The respective line 264 has been improved with the replacement of “projecting” with “prominent”.
“The prominent role is required from regulators”
Comment 7: Line 298 "to at their level" needs deleting
Answer: Thanks for your valuable comments. Line 298 has been improved.
“These authorities should develop policies for all levels of the drug supply chain, also support the manufacturers for local production of anti-cancer drugs and their raw materials, and other stakeholders financially to increase access.”
Reviewer 3 Report
Dear authors,
thank you so much for your efforts in scientific research. This is a very interesting and timely article.
As a minor addition, given the increasing importance of telepharmacy, in my humble opinion, I think that the use of telepharmacy could help in solving the problem of anti-cancer shortage and should be added as a possible solution. You can read for example Telepharmacy Services: Present Status and Future Perspectives: A Review DOI: 10.3390/medicine55070327
Thanks again for your contribution.
Best regards.
Author Response
Response to the Reviewer 3
Thanks for your valuable comment and suggestion.
Comment 1: As a minor addition, given the increasing importance of telepharmacy, in my humble opinion, I think that the use of telepharmacy could help in solving the problem of anti-cancer shortage and should be added as a possible solution. You can read for example Telepharmacy Services: Present Status and Future Perspectives: A Review DOI: 10.3390/medicine55070327
Answer: Thanks for your appreciation and valuable comment. Tele-pharmacy is widely used to provide pharmaceutical services to underserved areas and to address the problem of pharmacist shortage. For the problem of shortages of anti-cancer medicines, maybe tele-pharmacy can be used in the process of choosing alternatives. In the discussion, we added:
“……tele-pharmacy also can contribute to those hospitals’ lack of competent pharmacists in the aspect of choosing suitable alternative medicines”
Reviewer 4 Report
Dear authors,
This study, which investigated the shortage of anti-cancer drugs in Pakistan, is very interesting.
Study design and data collection were performed appropriately.
Here are some things to improve your manuscript.
There is the possibility that the lack of drugs is permanent and not caused by the Covid 19 pandemic. Therefore, it must be specified for what period of time the lack of anticancer medicines was studied (example: 10 years, 5 years, 1 year, during the Covid 19 pandemic, etc.). It is necessary to specify whether the lack of anticancer drugs is due to the Covid 19 pandemic.
The study has some limitations. First, oncologists should also participate in this study. Second, it would be very interesting to interview cancer patients.
I wish you good luck!
Author Response
Response to the Reviewer 4
Thanks for your valuable comments and suggestions.
Comment 1: There is the possibility that the lack of drugs is permanent and not caused by the Covid 19 pandemic. Therefore, it must be specified for what period of time the lack of anticancer medicines was studied (example: 10 years, 5 years, 1 year, during the Covid 19 pandemic, etc.). It is necessary to specify whether the lack of anticancer drugs is due to the Covid 19 pandemic.
Answer: Thanks for your appreciation, valuable comments, and suggestions.
The study was done after the COVID-19 pandemic from August to October 2021 when pharmacists expressed their experience with anti-cancer drug shortages. As Pakistan is importing most of its anti-cancer drugs, the lack of anti-cancer drugs is a current issue in Pakistan but COVID-19 aggravated this issue as mentioned by participant 8 in Table 2, “Theme 1: Dynamics and perception of anti-cancer medicines shortage.”
“In COVID-19, logistics are involved and major shortages happened. The import clearances of single brand/drug also take enough time. Drug is present at Pakistani port but not available to the patient. So before COVID-19, it was monthly but in COVID-19, it is daily or weekly (P8).”
Comment 2: The study has some limitations. First, oncologists should also participate in this study. Second, it would be very interesting to interview cancer patients.
Answer: Thanks for your valuable comment and suggestions. The above limitations are added in the limitation section and we also mentioned them for future studies.